# Characterization of Artisanal Spontaneous Sourdough Wheat Bread from Central Greece: Evaluation of Physico-Chemical, Microbiological, and Sensory Properties in Relation to Conventional Yeast Leavened Wheat Bread

**DOI:** 10.3390/foods10030635

**Published:** 2021-03-17

**Authors:** Pavlina Katsi, Ioanna S. Kosma, Sofia Michailidou, Anagnostis Argiriou, Anastasia V. Badeka, Michael G. Kontominas

**Affiliations:** 1Laboratory of Food Chemistry, Department of Chemistry, University of Ioannina, 45110 Ioannina, Greece; pavlina_katsi@hotmail.com (P.K.); i.kosma@uoi.gr (I.S.K.); 2Centre for Research and Technology Hellas, Institute of Applied Biosciences, 6th km Charilaou-Thermis, 57001 Thessaloniki, Greece; sofia_micha28@certh.gr (S.M.); argiriou@certh.gr (A.A.); 3Department of Food Science and Nutrition, University of the Aegean, 81400 Myrina, Lemnos, Greece

**Keywords:** sourdough bread, yeast leavened bread, microbiological, physico-chemical, sensory analysis, shelf life

## Abstract

In the present study, both yeast leavened bread (YLB) and artisanal sourdough wheat bread (SDB) were prepared. The physico-chemical, microbiological, and sensory properties of breads were monitored as a function of storage time (T = 25 °C). As expected, the titratable acidity (TA) values of SDB were higher than those of YLB. The aroma profile of SDB was similar to that of YLB, including classes of compounds such as alcohols, aldehydes, ketones, esters, organic acids, terpenes, and sulfur compounds; however, the concentrations between the two were different. Aroma deterioration of bread during storage was partly related to the loss of several volatiles. Texture and sensory analysis showed that SDB was harder, less elastic, but richer in aroma and light sour taste than YLB. Mold growth was apparent when the population of yeasts/molds reached approximately 4 log cfu/g. This yeast/mold count was reached on days 4–5 for YLB and day 18 + for SDB. A 16S amplicon meta-barcoding analysis showed that the bacterial profile of SDB was dominated by a single genus, (*Lactobacillus).* Analysis of the eukaryotic load showed that at the genus level, *Saccharomyces* and *Alternaria* were the most abundant genera, independently of the gene sequenced (18S or ITS). Based primarily on mold growth and texture data, which proved to be the most sensitive quality parameters, the shelf life was ca. 4–5 days for YLB and 10 days for SDB.

## 1. Introduction

Bread is probably the oldest “processed” food product. Carved images of bread furnaces have been recovered from Memphis ruins in Egypt as soon as 3000 B.C. Ancient Egyptians had over 50 different types of cakes, unleavened breads, and bread leavened with beer foam or sourdough [1]. Industrial wheat bread production started at the beginning of the 20th century after the introduction of baker’s yeast as a superior leavening agent to sourdough and brewer’s yeast. Ever since, the use of sourdough in bread production has been largely abandoned until very recently when consumer demand for the consumption of more natural bakery products with improved sensory properties and an extended shelf life, has revived the bakery industry’s interest in using sourdough for breadmaking.

According to Ganzle and Gobbetti [2], the advantages of using sourdough in bread making include acidification, improvement of dough properties, improvement of texture, flavor, leavening capacity, delayed staling, and increased resistance to microbial spoilage. Furthermore, sourdough enhances the nutritional value of bread through increasing mineral bioavailability, reducing the phytate content, lowering the postprandial glucose level, and providing certain exopolysaccharides with prebiotic and anti-staling properties. All the above benefits have been attributed to the lactic acid bacteria and yeasts naturally present in sourdough [3]. The sourdough lactic acid bacteria (LAB) fermentation creates an optimum pH for the activity of the endogenous enzymes (amylases and proteases) and improves bread loaf volume, delays starch retrogradation and bread firming, inhibits ropiness by spore-forming bacteria, and enhances flavor [4].

Sourdough is a mixture of flour and water that is naturally fermented with a symbiotic culture of LAB and yeasts, native to the flour itself, in a multi-day procedure to reach a pH value below 4.5. The levels of LAB in sourdough are 10^8^–10^9^ cfu/g, and the LAB/yeast ratio is generally 100:1 [5]. The most common LAB species found in sourdoughs are *Lb. acidophilus*, *Lb. farciminis*, *Lb. delbrueckii* (obligate homofermentative), *Lb. casei*, *Lb. plantarum*, *Lb. rhamnosus* (facultative heterofermentative), *Lb. brevis*, *Lb. sanfransicencis*, and *Lb. fermentum* (obligate heterofermentative) [6]. The yeast flora of sourdough is more homogenous. Universal sourdough yeasts appear to be *Saccharomyces cerevisiae, Candida milleri*, or *Candida humilis* [7]. LAB are mainly responsible for dough acidification and the modification of dough properties, while yeasts are mainly responsible for the production of flavor compounds.

The quality of sourdough bread is influenced by the specific microbiota developed in sourdough (starter cultures of LAB and yeasts), flour type (wheat/rye, flour extraction rate), flour/water ratio (dough yield), and process parameters such as initial pH, quantity of sourdough incorporated in dough, time and temperature of fermentation, etc. [5,8]. Specifically, temperature has a substantial impact on the dynamics of the microbial population and the metabolic activity of the microorganisms during fermentation. Homofermentative and facultative heterofermentative lactobacilli such as *Lb. fermentum* and *Lb. plantarum* predominate when the fermentation temperature is above 30 °C. In contrast, heterofermentative lactobacilli such as *Lb. sanfranciscensis* predominate when the temperature is lower than 30 °C [8]. 

The shelf life of bread (time period within which bread retains its acceptable quality and safety characteristics) is substantially limited due to several deterioration factors, mostly microbial spoilage and texture firming [9]. Fungal growth is the most common path of bread spoilage owed to species of *Penicillium* and *Aspergillus genera*. Bread can also be spoiled from bacteria due to high levels of moisture (*Bacillus subtilis, Bacillus cereus, Bacillus licheniformis*). Likewise, ropy bread is caused by *Bacillus subtilis*, resulting in the deterioration of bread texture [10]. Microbial spoilage of bread is due to cross-contamination post baking, as all microbiota is destroyed during the baking process [11].

Physico-chemical changes in bread may occur involving texture and flavor deterioration, rendering it stale. Staling is characterized by crumb firming mainly due to the retrogradation of the starch polymers and interactions between starch and proteins, crust softening due to the transfer of moisture from the crumb to crust, and finally flavor changes. Such changes are responsible for the disposal of large quantities of bread (8–10%), resulting in considerable economical losses [12,13,14]. 

Based on the above, the objectives of this study were (i) to characterize bread prepared using baker’s yeast and artisanal sourdough on a qualitative and quantitative basis using physico-chemical, microbiological, and sensory analyses and (ii) to identify the specific microbiota dominating the sourdough bread.

## 2. Materials and Methods 

### 2.1. Artisanal Sourdough Preparation

Two hundred grams of commercial wheat flour (85% extraction rate) produced by steel roller grinding) at St. George Mills S.A. Ioannina, Greece, were mixed with 2 g of table salt (0.6% *w/w*) and 200 mL of warm water (30 °C) in a bowl. The specific flour with a high extraction rate including a large part of the wheat bran is used to prepare sourdough in most parts of Greece. The ingredients were mixed using a sterilized spatula until a homogeneous thick batter was formed. Then, the bowl was covered with a hand towel and left overnight to rest at a temperature of 25 ± 1 °C. The following day, the dough was kneaded once more after adding 50 g flour and warm water (30 °C). This procedure was repeated for an additional 4 days, at which point the dough had matured/risen, forming a large number of holes in its mass and surface, which is a sign of vigorous fermentation activity, and giving off a sour and fresh odor. The mature dough, referred to as “sourdough”, was placed in a mason-type jar and stored in the refrigerator while a part of it was used for bread making. The particular recipe for the preparation of the sourdough was that used in the area of Northern Nafpaktia, central Greece.

### 2.2. Bread Preparation

Two types of bread were prepared; bread leavened with baker’s yeast (control, YLB) and sourdough bread (SDB). Control bread was prepared using 1000 g flour (1/3 wheat flour, 70% extraction rate, and 2/3 whole wheat flour), 700 mL of water, 6 g of salt (0.6% *w/w*) and 4 g of commercial baker’s yeast (GIOTIS S.A., Athens, Greece) added to the flour mix. The specific flour mixture was used to simulate the flour with an 85% extraction rate used for the preparation of sourdough, given the fact that baker’s yeast works better with flour with a 70% extraction rate. All above three types of flour used (70, 85, 100% extraction rate) are of commercial grade and were produced at St. George Mills, Ioannina, Greece. Then, the mixture was kneaded for 20 min, and the dough was left to rest at 25 °C for 50 min. Then, the mature dough was baked in a preheated home-type stove oven at 175 °C for 90 min. The resulting bread loaf weighed 1300 g. For the preparation of sourdough bread, to the above flour mixture, 200 g of sourdough, 6g salt, and water were added, and the mixture was kneaded for 20 min. The dough formed was left to rest at 25 °C for 5 h 15 min. Then, the mature dough was baked as stated for the control bread. Bread loaves, five for each type of bread of parallelepiped shape, were stored in a bread basket away from direct light, at 25 °C/60% Relative Humidity (RH). Sampling of bread for all analyses was carried out every other day for up to 18 days (days 0, 2, 4, 6, 8, 10 12, 14, 16, and 18). Both YLB and SDB prepared were reduced sodium breads [15]. Bread loaves were sliced (3 cm thick), and all were samples collected came from the inner parts of the loaf. The first sampling was carried out 1 h after the removal of bread from the oven.

### 2.3. Microbiological Analysis

#### 2.3.1. Bread Microflora

The following groups of microbiota were determined in breads according to official protocols [16]. Total viable counts (TVC), yeasts/molds, *Bacillus cereus*, LAB, and Enterobacteriaceae. All plates were examined visually for typical colony types and morphological characteristics associated with each growth medium. In addition, the selectivity of each medium was checked routinely by Gram staining and microscopic examination of smears prepared from randomly selected colonies from all of the media. TVC was determined using Plate Count Agar (PCA, Merck, Darmstadt, Germany). Colonies were counted after incubation at 30 °C for 48 h. Yeasts/molds were determined using the non-selective medium Rose Bengal Chloramphenicol agar (RBC, Merck). Colonies were counted after incubation at 30 °C for 4 days. *Bacillus cereus* was determined using the selective medium Mannitol Egg Yolk Polymyxin Agar (MYP, Oxoid) containing 50 mL egg yolk emulsion and 2 ampules of Polymyxin B supplement. Colonies were counted after incubation at 37 °C after 3 days. LAB were determined using the selective medium Man Rogosa Sharpe medium (MRS, Merck). Colonies were counted after incubation at 37 °C after 3 days. Finally, for members of the family Enterobacteriaceae, 1.0 mL sample was inoculated into 15 mL of molten (45 °C) violet red bile glucose agar (Oxoid). After setting, a 10 mL overlay of molten medium was added, and incubation was carried out at 37 °C for 24 h. 

#### 2.3.2. Sourdough Microbiota-Library Construction and Sequencing

DNA was extracted from 200 mg sourdough using the ZymoBIOMICS DNA Miniprep Kit (ZYMO RESEARCH; Irvine, CA, USA) according to the manufacturer’s instructions. DNA concentration was measured on a Qubit^TM^ 4 Fluorimeter using the Qubit^®^ dsDNA BR assay kit (Invitrogen, Carlsbad, CA, USA). Bacterial diversity was assessed by sequencing the V3–V4 hypervariable regions of the 16S rRNA gene. For fungal diversity, two approaches were applied; sequencing of the V7–V8 hypervariable regions of the 18S rRNA and application of the internal transcribed spacer 1 (ITS1) genes to assess fungal load. For the amplification of the 16S rRNA gene sequences, libraries were constructed using the primers D-Bact-0341-b-S-17 and D-Bact-0008-a-S-16 selected from Klindworth et al. [17]. For the amplification of the SSU of the 18S rRNA gene sequences, universal primers FR1 and FF390 were selected from Chemidlin Prevost-Boure et al. [18], whereas for ITS sequencing, primers BITS and B58S3 were selected from Bokulich and Mills [19]. All primers were modified by adding an Illumina (Illumina Inc, San Diego, CA, USA) overhang adapter nucleotide sequence at the 5′ end. Libraries were constructed as described in the Illumina’s 16S Metagenomic Sequencing Library Preparation (15044223 B) protocol. Libraries were quantified through quantitative PCR (qPCR) with the KAPA Library Quantification kit for Illumina sequencing platforms (KAPA BIOSYSTEMS, Woburn, MA, USA). Sequencing was performed in a MiSeq platform, using the MiSeq^®^ reagent kit v3 (2 × 300 cycles) (Illumina, San Diego, CA, USA). 

#### 2.3.3. Amplicon Meta-Barcoding Bioinformatics Analysis

Assessment of bacterial and fungal load was conducted using the Quantitative Insights into Microbial Ecology 2 (QIIME2) pipeline [20]. PCR primers were trimmed using the cutadapt plugin [21]. Trimming, de-noising, and chimera removal were conducted within QIIME2 with DADA2 R library [22]. Sequences were clustered into operational taxonomic units (OTUs) with 99% sequence similarity and aligned to the SILVA 132 database for 16S and 18S rRNA sequences. For ITS analysis, sequences were aligned to the UNITE fungal ITS database version 7.2. OTU tables and biom files were imported in R version 3.6.0 [23] to further process and visualize results. OTU counts and taxonomic assignments were merged to a phyloseq object with phyloseq R package [24]. All plots were visualized by combining functions provided by the ggplot2 R package [25]. All bar plots were normalized to 100% as abundance estimations within each sample; thus, percentages do not reflect the true biomass fraction of each sample. 

### 2.4. Physicochemical Analysis

#### 2.4.1. pH/Titratable Acidity (TA) Determination

pH and TA expressed as mL 0.1 N NaOH were determined in breads according to Latou et al. [26].

#### 2.4.2. Semi-Quantitative Determination of Flavor Volatiles of Bread 

Volatile compounds were identified and semi-quantified using the Solid Phase Micro Extraction GC/MS method as described by Latou et al. [26]. The sample consisted of 1 g of shredded bread (crumb and crust) and 20 μL of 4-methyl-2-pentanol (16.2 μg/kg) as an internal standard.

#### 2.4.3. Determination of Water Activity

Water activity (a_w_) of bread was determined using a Novasina model LabSwift-a_w_ water activity meter (Lachen, SZ) at 20 °C.

#### 2.4.4. Determination of Texture Parameters (Texture Profile Analysis, TPA) 

Objective texture profile analysis of breads was carried out using an Instron Universal Testing Machine, model 4411 (Instron Corp., Bucks, UK) providing compressive load vs. time curves. Bread crumb samples in the form of cylindrical discs (25 mm diameter × 20 mm in thickness) were prepared from the central part of each bread loaf. Then, samples were subjected to a double compression cycle using a 7.5 cm diameter stainless steel compression plunger. The crosshead speed was 100 mm/min, and the specimen compression depth was 60%. Texture parameters determined included hardness, springiness, cohesiveness, gumminess, and chewiness. Of the above parameters, the present study focused on hardness (defined as the peak force of the first compression cycle, expressed in N) and springiness (defined as the ratio of the time duration of force input during the second compression to that during the first compression, dimensionless) [27]. The Bluehill software (Version 1.4, Instron, Norwood, MA, U.S.A) was used to record texture profile analysis (TPA) data. 

### 2.5. Sensory Evaluation 

Sensory evaluation of breads (hedonic test) was conducted by a 51-member untrained panel consisting of graduate students of the Department of Chemistry according to Latou et al. [26]. Data on appearance, taste, aroma, and texture were collected every other day up to 18 days of storage (10 samplings in all). For testing, ca. 10 g of bread samples were placed in 3-digit randomly coded plastic cups and tightly capped. At each sampling day, along with the two test samples, panelists were given two reference samples consisting of bread taken from the same loaf (yeast leavened and sourdough bread) packed under N2 and stored in the dark at 2 °C up to 18 days. Prior to evaluation, samples were allowed to stand for 30 min in order to achieve the equilibration of volatile compounds in the cup headspace. Sensory scores were recorded on specially designed evaluation sheets using a 5-point hedonic scale where 5 = most liked and 1 = most disliked sample. The lower limit of acceptability for all four sensory attributes was the score of 3. On each sampling day, triplicate samples were evaluated for each treatment. If even one of the triplicate samples showed visible mold growth or if odor, taste, or texture received a score less than 3, the sample was considered unacceptable.

### 2.6. Statistical Analysis

The experiment was replicated twice on different occasions using different bread samples. Triplicate samples were analyzed per replicate (n = 2 × 3 = 6). Microbiological data were transformed into logarithms, expressed as log cfu/g), and subjected to analysis of variance using the software SPSS 16 for windows. Results are reported as mean values ± standard error or standard deviation. Tukey’s test was used to assess differences between means and the significance of differences was considered at the level of *p* < 0.05.

## 3. Results and Discussion

### 3.1. Microbiological Analysis

#### 3.1.1. Bread Microflora

Bread is considered a sterile product being heated at a high temperature (>170 °C) during baking. Thus, the presence of microbiota in bread is mostly due to post-baking contamination. The initial TVC for yeast leavened bread and sourdough bread was 2.9 and <2.00 log cfu/g, respectively (*p* < 0.05) (Figure 1a). TVC exceeded the value of 6.0 log cfu/g, which was considered as the upper microbiological limit for bakery products, as defined by the International Commission on Microbiological Specifications for Foods (ICMSF) [28] on day 4 for conventional bread. In traditional sourdough bread, TVC increased slowly but remained lower than 3.5 log cfu/g until day 18 of storage. It is obvious that TVC in sourdough bread remained quite lower (*p* < 0.05) than that of yeast leavened bread due to the substantially higher titratable acidity (7.9–8.6 mL 0.1 N NaOH) and lower pH (pH < 6, Figure 4b) of the former creating an effective hurdle to bacterial growth. 

Mohsen et al. [4] investigated the quality characteristics of Egyptian balady bread by using sourdough containing (2% *Saccharomyces cerevisiae* plus 1, 2, or 3% *Lactobacillus plantarum*). TVC, LAB, yeasts, pH, organic acids, and antimicrobial activity were evaluated during sourdough fermentation. Results showed an increase in organic acids, antimicrobial activity, and reduction in pH during the preparation of different sourdough samples. Bread characteristics showed an extented shelf life of 8 days for bread samples containing sourdough (2 or 3% *Lb. plantarum*) compared to 3 days for control bread. Improvements in the sensory characteristics and acceptability of balady bread were also recorded. The addition of 20% sourdough containing 2 or 3% *Lb. plantarum* to wheat flour dough also retarded the staling rate compared to control samples. Latou et al. [26] studied the effect of active packaging (addition of ethanol emitter/oxygen absorber) on shelf-life extension of sliced wheat bread stored at 20 °C. The TVC of control samples ranged between 3.0 and 7.2 log cfu/g after 10 days of storage. These values are in excellent agreement with those of the present study regarding yeast leavened bread. Jonkuvienė et al. [29] identified LAB naturally occurring in spontaneous sourdough and used them for quality improvement and prolonging the shelf life of rye, wheat + rye, and wheat bread. The identification of isolates from spontaneous sourdough by the pyrosequencing assay showed that *Lactobacillus reuteri* was the dominant lactic acid bacterium. *Lb. reuteri* showed a high preserving capacity also against fungi during storage exhibiting high antimicrobial activity. The authors concluded that *Lb. reuteri* has the potential to be used as a starter additive that could improve safety and enhance the shelf life of bread. 

The initial count of yeasts/molds was lower than 2.0 log cfu/g for both bread types reaching values of 6.8 and 3.7 log cfu/g after 8 days for the control and 18 days for the sourdough bread (Figure 1b). The yeast/mold growth rate rose sharply beginning with days 2–3 of storage for yeast bread, while a considerably lower (*p* < 0.05) yeast/mold growth rate was observed beginning with days 10–12 for sourdough bread. The explanation in this trend is similar to that for TVC—that is, the lower pH/higher titratable acidity of the sourdough bread inhibited the rapid growth of yeasts/molds. It should be noted that mold growth was apparent when the population of yeasts/molds reached approximately 4 log cfu/g. This observation is in agreement with Latou et al. [26]. This yeasts/molds count was reached on days 4–5 for the control and day 18+ for the sourdough bread. It is clear that the sourdough resulted in a drastic reduction of yeasts/molds populations. Fernandez et al. [30] reported a count of 3.0 log cfu/g for yeasts/molds’ visible growth in soy bread. Finally, Tatar et al. [31] prepared wheat bread with the addition of different levels of acidulants, i.e., acetic and lactic acid with calcium propionate. The study included microbiological analysis, sensory evaluation, and shelf-life determination. Storage for 4 days affected the aroma, taste, texture, and bread crust characteristics to a great extent. Treatments containing 0.2% and 0.3% lactic acid in combination with 0.2% calcium propionate were the most effective against microbial spoilage. TVC on day 4 ranged from 2.5 to 3.1 log cfu/g, while molds ranged from 3 to 3.5 log cfu/g. According to Samapundo et al. [32], fungal spoilage is a minor problem in the bakery industry today but has been causing trouble and losses for decades.

*B. cereus* is a Gram-positive, spore-forming, motile, facultative anaerobic bacterium causing both food spoilage in bread (ropy texture) and possibly food poisoning [33]. On day 0, *B. cereus* counts were under the method detection limit (2 log cfu/g), and at the time of visible signs of mold growth (days 4–5), they reached 5.8 log cfu/g for control samples and 2.1 log cfu/g for sourdough bread samples (*p* < 0.05) (Figure 1c). According to European Food Safety Authority (EFSA) [34], food-borne diseases caused by *B. cereus* have been associated with concentrations of at least 5 log cfu/g, which is a value that was exceeded in the present study only for the control bread sample on days 4–5 of storage. *B. cereus* population reached 6.3 log cfu/g in the control sample on day 8 of storage and 3.5 log cfu/g in sourdough bread on day 18 of storage. It is obvious that sourdough inhibits the growth of *B. cereus* due to the increased acidity of the bread matrix acting as a hurdle for bacterial growth. Latou et al. [26] reported a *B. cereus* count of 6 log cfu/g for conventional air packaged bread on day 9 of storage. At the time of visible mold growth (day 5 of storage), the population of *B. cereus* was ca. 4 log cfu/g. Bailey and von Holy [35] studied the contamination of bread by Bacillus spp. and reported enhanced growth (6.4 log cfu/g) on day 3 of storage at 30 °C. Differences in *B. cereus* populations between the research cited above and the present study are probably due to the lower storage temperature used in the present study (25 vs. 30 °C). Ravimannan [36] investigated the microbial profile of bread in Sri Lanka stored for 5 days at 29–31 °C and 70% relative humidity. After the initial fungal growth (mainly *Mucor spp.* and *Rhizopus spp*. after three days), the soft texture of the bread changed and became hard, during which the microbial load was dominated by bacteria. The bacterial species identified were *B. cereus* and *B. subtilis.* On day 3 of storage, the combined population of *B. cereus* and *B. subtilis* was 5.1 log cfu/g, while on day 5 of storage, the respective population was 6.3 log cfu/g. It was suggested by the authors that bread should be consumed during the first three days after preparation.

LAB are fermentative, facultative anaerobic bacteria that can grow both in the presence and absence of oxygen. They constitute the major part of sourdough microbiota and the major source of sourdough bread acidity. The initial LAB counts were lower than 1 log cfu/g and reached ca. 5.2 log cfu/g in the control sample on day 6 of storage (Figure 1d). On the same day, LAB counts remained under 1 log cfu/g in the sourdough bread (*p* < 0.05). Starting with day 12 of storage, the LAB count began to increase in sourdough bread, reaching ca. 3.4 log cfu/g on day 18. The large difference in LAB counts between yeast leavened and sourdough bread may be justified by the respective higher pH values in the former affecting LAB growth. Except for certain *Lactobacillus* spp., lactic acid bacteria are probably best characterized as neutrophiles with optimal growth rates ranging between 6.3 and 6.9 [37]. Park et al. [38] studied the LAB growth in Korean jeung-pyun (sponge-like sourdough bread cooked in steam) prepared by fermenting rice flour with rice wine. The study showed that LAB in sourdough (6–7 log cfu/g) play a key role in production of jeung-pyun bread, influencing the product textural and sensory properties. 

Enterobacteriaceae comprises a large group of facultative anaerobic bacteria that are widely used as indicators of hygiene conditions prevailing in the food processing environment. The enterobacteriaceae remained below 1 log cfu/g for all bread treatments during the 18 days of storage (data not shown). According to the New South Wales (NSW) Australian food authority, a count of Enterobacteriaceae <10^2^ cfu/g, is considered as good, 10^2^ to < 10^4^ cfu/g is considered acceptable, and ≥10^4^ cfu/g is regarded as unsatisfactory in ready-to-eat foods [39]. Both breads in the present study met this good quality criterion for the entire storage period. 

Latou et al. [26] reported the same finding for sliced what bread stored both aerobically and under various active packaging conditions.

#### 3.1.2. Microbial Community in Sourdough

The microbial community in sourdough was assessed using meta-barcoding analysis both for prokaryotic (16S rRNA) and eukaryotic organisms (18S rRNA and ITS genes). Sequencing of the 16S rRNA gene resulted in 107.167 raw reads; after quality filtering and the removal of chimeric sequences, 70.145 reads were obtained. For eukaryotes, the sequencing of 18S rRNA and ITS genes resulted in 143.801 and 76.328 raw reads, respectively. After quality filtering and the removal of chimeric reads, these numbers were reduced to 88.189 and 14.139 reads, respectively. The assignment and clustering of sequences against the reference databases resulted in 61 unique OTUs for bacteria, 13 unique OTUs derived from 18S rRNA sequencing, and 75 unique OTUs derived from ITS sequencing (Table 1). 

The bacterial profile of sourdough revealed that it was dominated by Firmicutes (89.9%) followed by Proteobacteria (7.05%) and Cyanobacteria (2.86%) (Figure 2a). Bacilli (89.91%), Alphaproteobacteria (6.94%), and Oxyphotobacteria (2.86%) dominated the bacterial community at class level (Figure 2b). Overall, sourdough presented low bacterial diversity; genus *Lactobacillus* (89.91%), mainly represented by *L. sanfranciscensis* spp. (89.67%) was the dominant bacteria, followed by *L. paralimentarius* (0.28%) (Figure 2d,e).

Analysis of the eukaryotic load identified a more complex profile for the sourdough, independently of the target region (18S rRNA or ITS genes). Yet, analysis of the ITS gene identified more OTUs, which are better characterized at the species level. Analysis of the 18S rRNA gene revealed that Opisthokonta was the major phylum being present with 75.73% relative abundance (data not shown). These fungi were further classified at the family level; Saccharomycetaceae (33.03%), Pleosporaceae (17.52%), and Cladosporiaceae (11.86%) were the most abundant families present in sourdough (Figure 3a). At the genus level, 18S rRNA analysis revealed that *Saccharomyces* (33.03%) was the dominant genus, followed by *Alternaria* (12.22%), *Cladosporium* (11.86%), and *Fusarium* (1.82%) genera (Figure 3b). At the species level, 18S rRNA analysis against SILVA 132 could not detect any fungi. Although 18S rRNA analysis identified the main bacteria at the genus level, analysis of the ITS gene provided much more information on the fungal load of sourdough. In particular, Saccharomycetaceae (28.81%), Pleosporaceae (21.93%), and Mycosphaerellaceae (2.23%) were the most abundant families (Figure 3c). At the genus level, *Saccharomyces* (28.81%) and *Alternaria* (20.31%) were the most abundant genera, followed by *Mycosphaerella* (2.23%) (Figure 3d). These genera were mainly represented in sourdough by *S. cerevisiae* (28.81%), *A. hordeicola* (5.88%), and *M. tassiana* (2.23%), respectively (Figure 3e). It is noteworthy that a large fraction of the eukaryotic load (41.25%) could not be identified at the genus level. To the best of our knowledge *A. hordeicola and M. tassiana* have not been previously reported in traditional sourdoughs.

The above results are in general agreement with those of De Vuyst et al. [40], who isolated bacteria into the species *Lactobacillus sanfranciscensis*, *Lactobacillus brevis*, *Lactobacillus paralimentarius*, and *Weissella cibaria*. This consortium according to these authors seems to be unique for the Greek traditional wheat sourdoughs studied. Strains of the species *W. cibaria* have not been isolated from sourdoughs previously. Studies dealing with the identification and characterization of LAB from traditional sourdoughs revealed the dominance of *L. sanfranciscensis* strains in type I sourdoughs, which were probably selected only by the environmental conditions induced by the sourdough fermentation technology. However, many researchers still report the existence of unidentifiable and perhaps new sourdough LAB species. Palla et al. [41] characterized sourdough microbiota used to produce Protected Designation of Origin (PDO) Tuscan bread and reported a large number of *L. sanfranciscensis* and *C. milleri* strains, along with a few of *S. cerevisiae*. Lhomme et al. [42] characterized 16 sourdoughs used for the manufacture of traditional French breads. *Lactobacillus sanfranciscensis* was found to be the dominant species in French sourdoughs. In addition to species frequently encountered (e.g., *Lactobacillus parabrevis/Lactobacillus hammesii, Lactobacillus plantarum*, and *Leuconostoc mesenteroides*), *Lactobacillus xiangfangensis* and *Lactobacillus diolivorans* were also found in sourdough. Yeast diversity was lower than LAB. Except for one sourdough (solely dominated by *Kazachstania servazzii*)*,* the yeast microbiota of French sourdoughs was dominated by *Saccharomyces cerevisiae*. Finally, according to De Vuyst et al. [8], no clear-cut relationship between a typical sourdough and its associated microbiota can be found, as this is dependent on the sampling, isolation, and identification procedures. Both simple and very complex consortia may be encountered. Moreover, intrinsic and extrinsic factors may influence the composition of the sourdough microbiota such as the flour (type, quality status, etc.) and the process parameters (temperature, pH, dough yield, backslopping practices, etc.)

### 3.2. Physico-Chemical Analysis

#### 3.2.1. TA/pH/a_w_

The initial TA of sourdough bread was 7.9, increasing to 8.7 mL NaOH 0.1 N on day 18 of storage (Figure 4a). Initial TA values for the yeast leavened bread were 4.0, increasing to 6.7 mL NaOH 0.1 N on day 8 of storage. On day 0, the pH of yeast leavened bread was 6.8, decreasing to 6.4 on day 8 of storage. The initial pH value of sourdough bread was 5.7, remaining practically constant and decreasing to 5.5 on day 18 of storage (Figure 4b). The results regarding TA are in line with the maximum growth of acid-producing LAB after days 2–3 for yeast leavened bread and after day 12 for sourdough bread (Figure 1d). The results regarding pH are in agreement with both LAB growth curves (Figure 1d) and TA curves (Figure 4a); i.e., pH values begin to decrease (*p* < 0.05) after ca. day 3 of storage (with a simultaneous increase in TA) for the yeast leavened bread and begin to decrease (*p* < 0.05) after days 14–15 of storage (with a simultaneous increase TA) for the sourdough bread.

Banu et al. [43] prepared rye bread with 20% sourdough using *Lb. plantarum* and *Lb. brevis*. Rye bread without sourdough was taken as the control. The sourdough rye bread had a lower pH and a higher TA (pH from 5.1 to 5.3, and TA from 5 to 5.4 mL NaOH 0.1 N), in comparison to control rye bread sample (pH 6.1, and TA 2.5 mL NaOH 0.1 N). Mert et al. [44] investigated the effect of the addition of different amounts of sourdough (0%, 20%, and 40% of flour basis) on the quality parameters (pH, TA, firmness, and volume) of chestnut-rice gluten-free breads. pH and TA of the control was 6.4 and 3.0 mL NaOH 0.1 N, respectively. pH and TA were 5.4 and 5.2 mL NaOH 0.1 N, respectively, for bread made of 20% sourdough. The respective values for bread made of 40% sourdough were pH = 4.9 and TA = 7.0 mL NaOH 0.1N. Likewise, Sanz-Penella et al. [45] used *Bifidibacterium pseudocatenulatum* as a starter culture in the preparation of sourdough, which was added at different ratios (0–20%) to whole wheat flour to prepare bread. The pH/TA values of the control bread were 5.7/4.2 mL NaOH 0.1 N. Respective values for the pH of sourdough breads ranged between 5.0 and 5.5 and for TA between 5.7 and 10.6 mL NaOH 0.1 N. Finally, Marcus et al. [46] reported a pH = 6.0 and TA = 2.8 mL NaOH 0.1 N for control bread and a pH = 5.0 and TA = 5.0 mL NaOH 0.1 N for sourdough bread prepared with the antifungal strain *Lactobacillus amylovorus* containing 28.3% of sourdough. Differences between the pH/TA values of the present study and the literature may be related to the type of sourdough culture used, ratio of sourdough/flour, temperature of sourdough fermentation, type of flour used, etc.

Water activity values for control and sourdough bread were 0.983 and 0.981, respectively. The lower a_w_ values (*p* < 0.05) in sourdough bread reflect the longer product shelf life compared to yeast leavened bread. Marcus et al. [46] reported a_w_ values of 0.982 for control bread and 0.980 for sourdough bread at a NaCl content of 0.6%. Respective values for a NaCl content of 1.2% were 0.974 and 0.972. NaCl is hygroscopic in nature, and therefore, the level of NaCl added to bread will have an impact on the water activity, which has previously been shown to affect microbial growth. Present a_w_ values are in agreement with those (0.96–0.98) published in the literature [47,48].

#### 3.2.2. Volatile Compounds

Reaction pathways for aroma formation in bread include volatiles formed (i) during fermentation (LAB and yeast activity), (ii) as a result of lipid oxidation, and (iii) from Maillard and caramelization reactions [49]. Furthermore, volatiles such as aldehydes and their corresponding alcohols are formed inside the yeast cells from the degradation of flour amino acids via the Ehrlich pathway [50]. Major volatiles formed during dough fermentation include alcohols, organic acids, esters, aldehydes, ketones, lactones, etc. mainly originating from precursors including carbohydrates and amino acids. The respective volatiles formed during lipid oxidation include aldehydes and ketones originating from the decomposition of triglycerides and fatty acids. Finally, volatiles formed from Maillard and caramelization reactions include pyrazines, pyridines, pyrroles, furans, sulfur compounds, aldehydes, and ketones originating from precursors including amino acids, peptides, and carbohydrates [3].

In total, 27 and 33 volatile compounds were determined in sourdough and yeast leavened bread, respectively. These belong to the chemical classes of aldehydes, ketones, alcohols, esters, organic acids, hydrocarbons, aromatic compounds, and sulfur compounds. The formation of these compounds as a function of time is given in (Table 2 and Table 3 and Figure 5). According to Hansen and Hansen [51], sourdough bread should have more volatile compounds than yeast leavened bread or at least higher volatile contents.

Alcohols identified included ethanol (alcoholic odor), 1-propanol (buttery odor), 2-methyl-1-propanol (ethereal odor), 3-methyl-1-butanol (malty odor), 2-methyl-1-butanol (fragrant odor), 1-hexanol (grass/sweet/woody odor), 1-octen-3-ol (acidic odor), and pentanol (fruity, sweet odor). The total alcohol concentration was higher (*p* < 0.05) in yeast bread compared to sourdough bread (Figure 5). Ethanol is the main product of yeast fermentation. Minor products of yeast fermentation include butanol and propanol derivatives [52]. Short-chain alcohols are produced through sugar fermentation, while long-chain alcohols are produced through amino acid metabolism [53]. Of the 33 sourdough LAB isolates studied in detail by De Vuyst et al. [40], all heterofermentative LAB strains produced ethanol, lactic acid, and acetic acid in agreement with the results of the present study. *L. sanfranciscensis* (see below) and *L. plantarum* produce a wide range of volatile compounds [51]. In addition to ethanol, lactic acid, and acetic acid, heterofermentative LAB also produce ethyl acetate with some alcohols and aldehydes.

Ketones and aldehydes identified in the present study include 3-hydroxy-2-butanone (buttery odor), 2,3-butanedione (intense buttery odor), 2-pentylfuranone (green/fruity odor), (hexanal (green/grass odor), 3-methyl-butanal and 2-methyl-butanal (malty odor), heptanal (green/fatty odor), benzaldehyde (bitter almond odor), octanal (fruity/soapy odor), nonanal (sweet/mellon odor), and furfural (roasted odor). Carbonyl compounds are mostly the products of hydroperoxide decomposition; i.e., hexanal and pentanal form through the decomposition of linoleic acid. The Maillard reaction also produces numerous aldehydes and ketones, i.e., 3-methyl-butanal being a product of the Maillard reaction [54] and ethanol, 2-methyl-1-propanol, 2-methyl-1-butanol, 3-methyl-1-butanol, and benzaldehyde being products of yeast fermentation [55]. LAB also liberate aroma precursors such as amino acids, which are degraded into aldehydes or the corresponding alcohols [55]. As shown in Figure 2, aldehyde concentration was higher (*p* < 0.05) in sourdough bread compared to yeast bread. The trend was reverse regarding ketones. 

Esters identified included methyl formate (ethereal odor), ethyl acetate (sweet odor), and methyl butyrate (apple–pineapple fruity odor). Ester production is mainly due to heterofermentative LAB metabolism [56]. Ester concentration was higher (*p* < 0.05) in yeast leavened compared to sourdough bread. Of the organic acids, acetic acid (pungent/sour odor) was identified only in sourdough bread at high concentrations. Acetic acid is the product of lactic acid fermentation. Heterofermentative LAB produce 50–65% lactic acid but also acetic acid, ethanol, and CO_2_ [57]. Of the hydrocarbons, hexane and other alkanes (alkane-like odor) were identified. 2-pentylfuran (fruity odor), and dimethyl-disulfide are products of LAB fermentation [56]. p-Cymene (mild pleasant odor) has been also identified by Seitz et al. [58] in sourdough bread. Hydrocarbon concentration was higher (*p* < 0.05) in the yeast leavened bread compared to the sourdough bread. Lastly, aromatic compounds were higher (*p* < 0.05) in sourdough compared to yeast leavened bread.

According to De Vuyst et al. [40,59], spontaneous sourdoughs are generally specific to a region because wild LAB and yeast species prevail depend on local ecological factors. Temperature, time, and the number of backsloppings influence the yeast and LAB fermentations and lipid oxidation and hence the volatile profile of sourdough [3]. Present results regarding the flavor profile of yeast leavened bread are in good agreement with those of Latou et al. [26], Bianchi et al. [27], and Cho and Peterson [60], while those of the sourdough bread are in good agreement with those of Petel et al. [3] and Plessas et al. [61]. In addition to the above volatile compounds, one would expect the presence of pyrazines [62], as bread involves baking. However, such compounds, were not identified in the present study. This may be attributed to the fact that pyrazines rapidly evaporate from the sample due to their high volatility.

A decrease in ethanol concentration after 5 days of storage of sourdough bread was also reported by Plessas et al. [61], which is in agreement with the present study. As a general rule, the deterioration of bread aroma may be the result of the loss of several volatiles during the bread ageing process as well as the formation of “off-flavors” occurring via the oxidation of bread lipids during storage. The above authors showed a dramatic decrease of the number and the amount of volatile compounds after five days of storage, which is something that is also observed in the present study. Soukoulis et al. [63] studied the volatile profile of bread prepared with the addition of lactic acid bacteria using Solid Phase Micro Extraction- Gas Chromatography/Mass Spectrometry (SPME-GC/MS) and identified most of the compounds present in sourdough bread found in the present study. Lastly, Xu et al. [64] used a mixture of starter cultures of yeast and lactobacilli (*L. sanfranciscensis* and *L. sakei*) and yeasts (*Kazachstania humilis*, *Saccharomyces cerevisiae*, and *Wickerhamomyces anomalus*) starter cultures to prepare sourdough wheat bread and studied the flavor volatiles produced. The volatile profile recorded was similar to that of the present study with the exception of a more complex ester profile compared to the present study.

#### 3.2.3. Mechanical Parameters (Texture Profile Analysis—TPA)

Of the mechanical properties determined using texture profile analysis (hardness, springiness, cohesiveness, gumminess, and chewiness), the present study focused on hardness and springiness (shown in Table 4a,b). Initial values of hardness were 4.06 ± 0.59 N for yeast bread and 7.15 ± 1.94 N for sourdough bread. Hardness increased with storage time (*p* < 0.05) in both breads. At the time of sensory rejection (day 4–5 for yeast bread and day 10 for sourdough bread), hardness reached 11.88 ± 3.42 N for yeast bread and 35.40 ± 5.89 N (*p* < 0.05) for sourdough bread. It is apparent that sourdough bread is harder in texture compared to yeast leavened bread. This finding is in agreement with Sanz-Penella et al. [45], who showed that sourdough bread hardness increases with amount of sourdough added to wheat flour. Likewise, Latou et al. [26] working with preservative-added wheat bread reported an initial (day 0) hardness value of 4.8 N, reaching 15.7 N on day 9 of storage for control bread. 

Springiness is a measure of bread samples to return to their initial condition upon compression. Initial values of springiness were 3.76 ± 0.92 N for yeast bread and 1.74 ± 0.39 N (*p* < 0.05) for sourdough bread. Springiness varied with storage time (*p* < 0.05) only for the sourdough bread. At the time of sensory rejection (days 5–6 for yeast bread and day 10 for sourdough bread), springiness reached 5.07 ± 0.70 N for yeast bread and 3.28 ± 0.30 N for sourdough bread.

It is clear that yeast bread is more elastic in texture compared to sourdough bread. These findings are in general agreement with those of Rinaldi et al. [65], who applied TPA to both yeast leavened and sourdough bread stored for 5 days. They reported that the sourdough bread was harder than the yeast bread and that the hardness increased in both breads with time. Cohesiveness was higher in the yeast bread in contrast to the results of the present study showing a higher cohesiveness for sourdough bread (results not shown). Casado et al. [66] ran TPA analysis in yeast and sourdough bread and reported a higher hardness and chewiness in sourdough bread. Springiness did not show statistically significant differences between the two breads. Finally, in contrast to the results of the present study, Hadaegh et al. [67] reported a lower hardness in toast wheat bread with the addition of sourdough compared to yeast leavened bread.

### 3.3. Sensory Analysis

Of the sensory attributes monitored (appearance, aroma, taste, and texture), aroma and taste showed a decreasing acceptability trend with time and proved equally sensitive sensory attributes for the quality evaluation of bread (Figure 6). Based on aroma and taste, yeast bread was rejected after 5–6 days, while sourdough bread was rejected after 10 days of storage. Similar results were recorded for texture. Finally, based on appearance, yeast bread was rejected after ca. 4–5 days, while sourdough bread was rejected after 16 days due to mold growth. The most sensitive sensory attribute for yeast leavened bread proved to be appearance (mold growth, day 5) as well as texture hardening (days 5–6), while for sourdough bread, the most sensitive sensory attribute was texture hardening (day 10). There was a partial agreement between sensory and microbiological data. Based on TVC, the shelf life of yeast leavened bread was 4–5 days (limit 10^6^ cfu/g) and 18+ days for sourdough bread stored at 25 °C. The same holds for the pathogen *B. cereus.* Mold growth limited the microbiological shelf life to day 5 for YLB and 16 days fir SDB. Finally, Enterobacteriaceae recorded very low counts (lower than 1 log cfu/g) for all treatments and did not critically affect product shelf life.

The sensory data of the present study are in good agreement with those in the literature. Mohsen et al. [4] conducted sensory evaluation on breads prepared with the addition of different amounts of LAB. They reported that the breads prepared with LAB received higher scores regarding texture, aroma, and taste compared to control breads. Edeghor et al. [68] prepared yeast leavened bread, bread containing LAB, and bread containing both cultures. The bread prepared with the addition of LAB received the highest scores regarding aroma and taste followed by the bread prepared with the mixed culture. Regarding appearance and texture, the bread with the mixed culture received the highest scores followed by the yeast leavened bread. Similar results were reported by Quintero Lira et al. [69], who compared bread prepared using baker’s yeast plus *Lb. paracasei* to that prepared using only baker’s yeast. Bread prepared with the mixed culture consistently received higher scores regarding appearance, texture, taste, and aroma. Marcus et al. [46] prepared (i) conventional bread, (ii) sourdough bread using the antifungal strain *Lactobacillus amylovorus*, and (iii) bread containing 0.5% calcium propionate (CA) commonly used in bread as an antifungal agent. For a salt content of 0.6%, they reported a shelf life of 2-3 days for control bread, 13–14 days for sourdough bread, and 12 days for bread containing (CA). Finally, the results of the present study are also in good agreement with those of Casado et al. [66] and Hadaegh et al. [67], who reported that the sensory panels preferred the sourdough bread samples to those of the yeast leaved bread.

## 4. Conclusions

In the present study, 16S analysis showed that the bacterial profile of sourdough was dominated by a single genus, (*Lactobacillus)*, whereas the eukaryotic load showed that at the genus level, *Saccharomyces and Alternaria* were the most abundant genera, independently of the gene sequenced (18S or ITS). *A. hordeicola and M. tassiana* were reported for the first time in the eukaryotic load of traditional sourdoughs. The use of sourdough in bread making resulted in a product that was harder and less elastic in texture, richer in aroma (higher concentration of organic acids, aromatic compounds, and aldehydes), and had a desirable light sour taste with a considerably longer shelf life compared to conventional yeast leavened bread. Based primarily on microbiological (mold growth), and sensory (texture) data, the shelf life of conventional yeast leavened bread was ca. 4–5 days and 10 days for artisanal sourdough bread stored at 25 °C. Depending on the specific microbiota associations in sourdough, type of flour used, and storage temperature of SDB, its quality characteristics and shelf life will vary. Most of the information in the literature on SDB stored at room temperature refers to a shelf life between 4 and 7 days. We have shown that using the specific sourdough, the shelf life of SDB is extended to 10 days even at slightly elevated ambient temperatures (25 °C). This finding along with the specific associations of LAB and yeasts in the artisanal sourdough used comprise the novelty of the present study.

## Figures and Tables

**Figure 1 foods-10-00635-f001:**
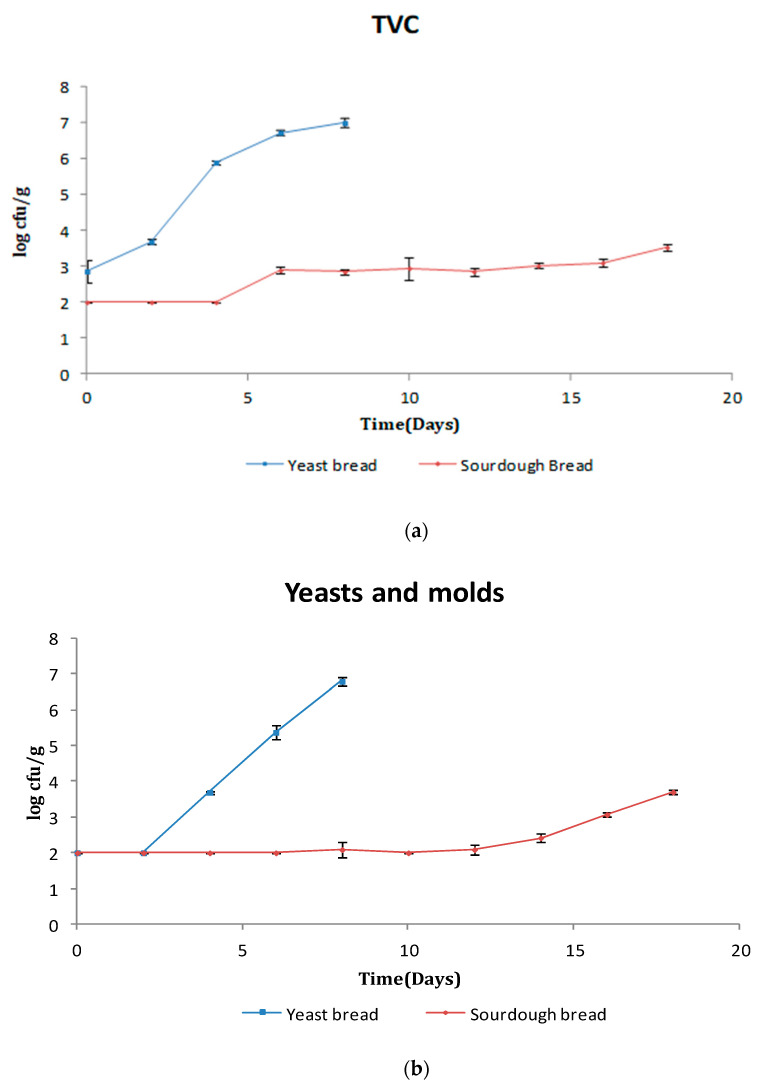
Changes in (**a**) total viable counts (TVC), (**b**) yeasts and molds, (**c**) *Bacillus cereus,* and (**d**) lactic acid bacteria (LAB) in yeast leavened and sourdough bread as a function of time.

**Figure 2 foods-10-00635-f002:**
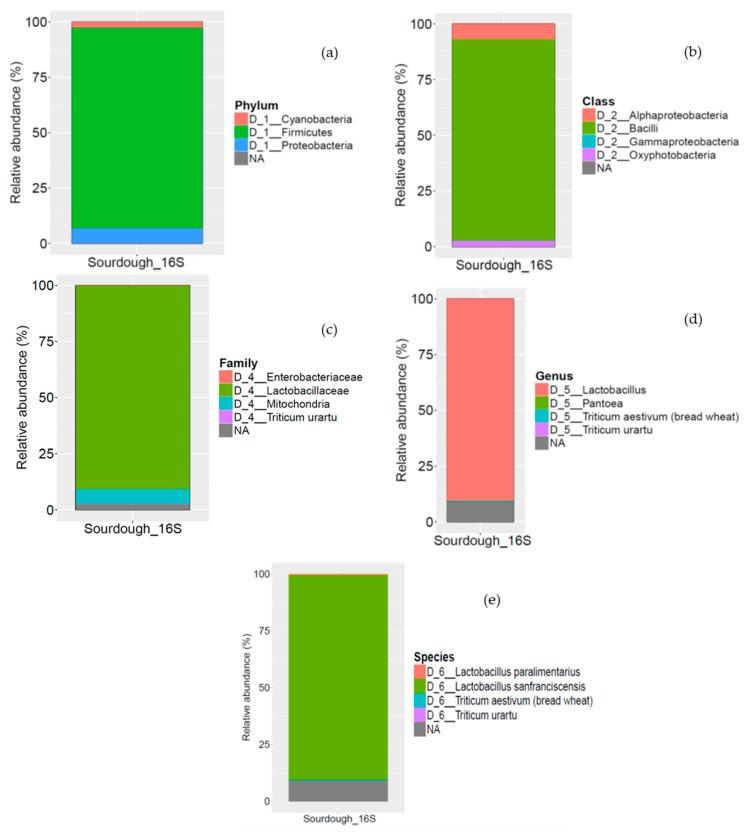
Analysis of prokaryotic community in sourdough (**a**), distribution of the major bacterial phyla, (**b**) distribution of the major bacteria at the class level, (**c**) distribution of the major bacteria at the family level, (**d**) distribution of the major bacteria at the genus level, and (**e**) distribution of the major bacteria at the species level. The scale in the y-axis reflects the normalized relative abundance percentages.

**Figure 3 foods-10-00635-f003:**
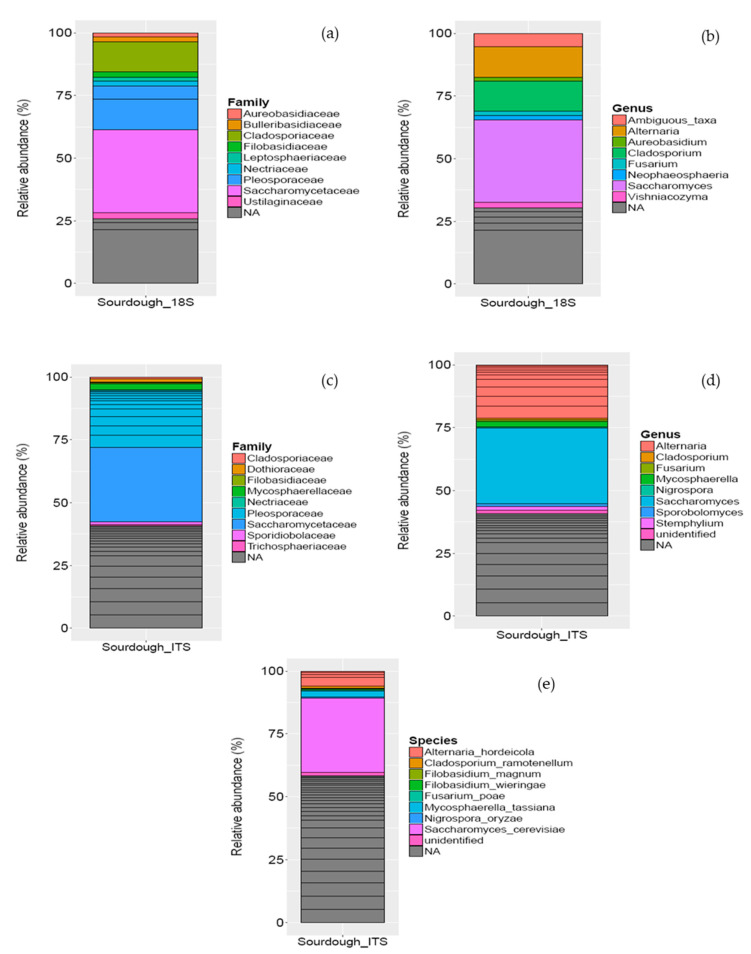
Analysis of the eukaryotic community in sourdough by sequencing the 18S rRNA gene (**a**) distribution of the major fungal families, (**b**) distribution of the major fungal genera. (**c**) Analysis of eukaryotic community in sourdough by sequencing the ITS gene. Distribution of the major fungal families, (**d**) distribution of the major eukaryotic genera, and (**e**) distribution of the major eukaryotic species. The scale in the y-axis reflects the normalized relative abundance percentages (%). Black lines within each bar separates each species into lower taxonomic levels.

**Figure 4 foods-10-00635-f004:**
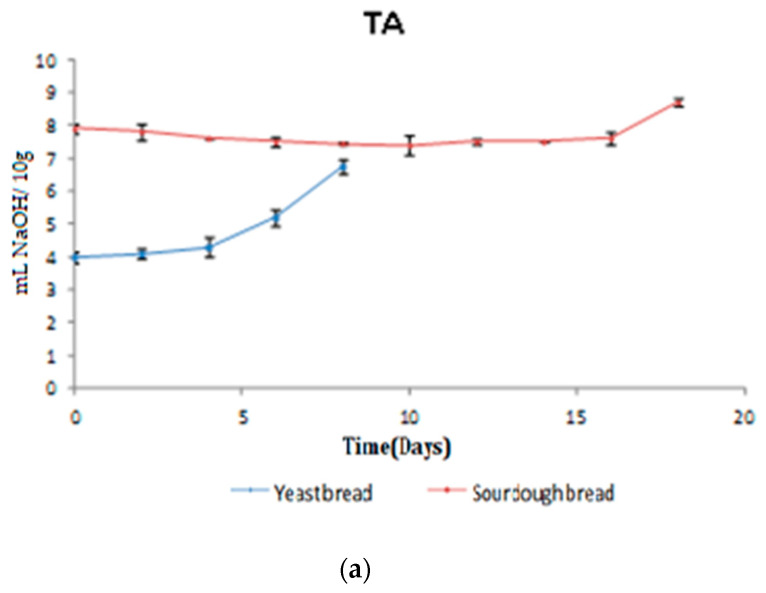
Changes in (**a**) titratable acidity and (**b**) pH in yeast leavened and sourdough bread as a function of time.

**Figure 5 foods-10-00635-f005:**
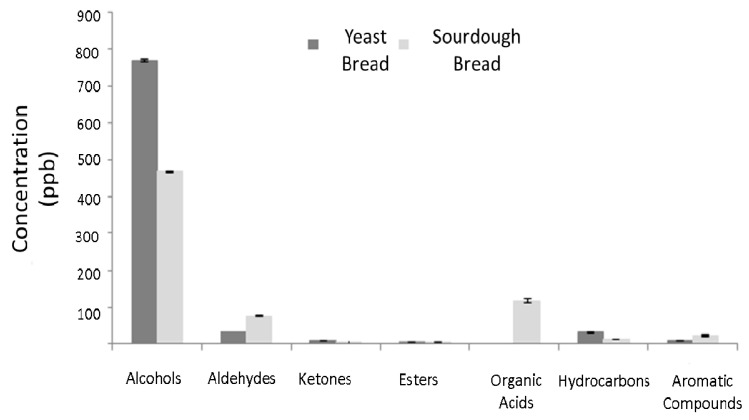
Bar diagram showing differences in volatile compounds between yeast bread and sourdough bread on day of bread preparation.

**Figure 6 foods-10-00635-f006:**
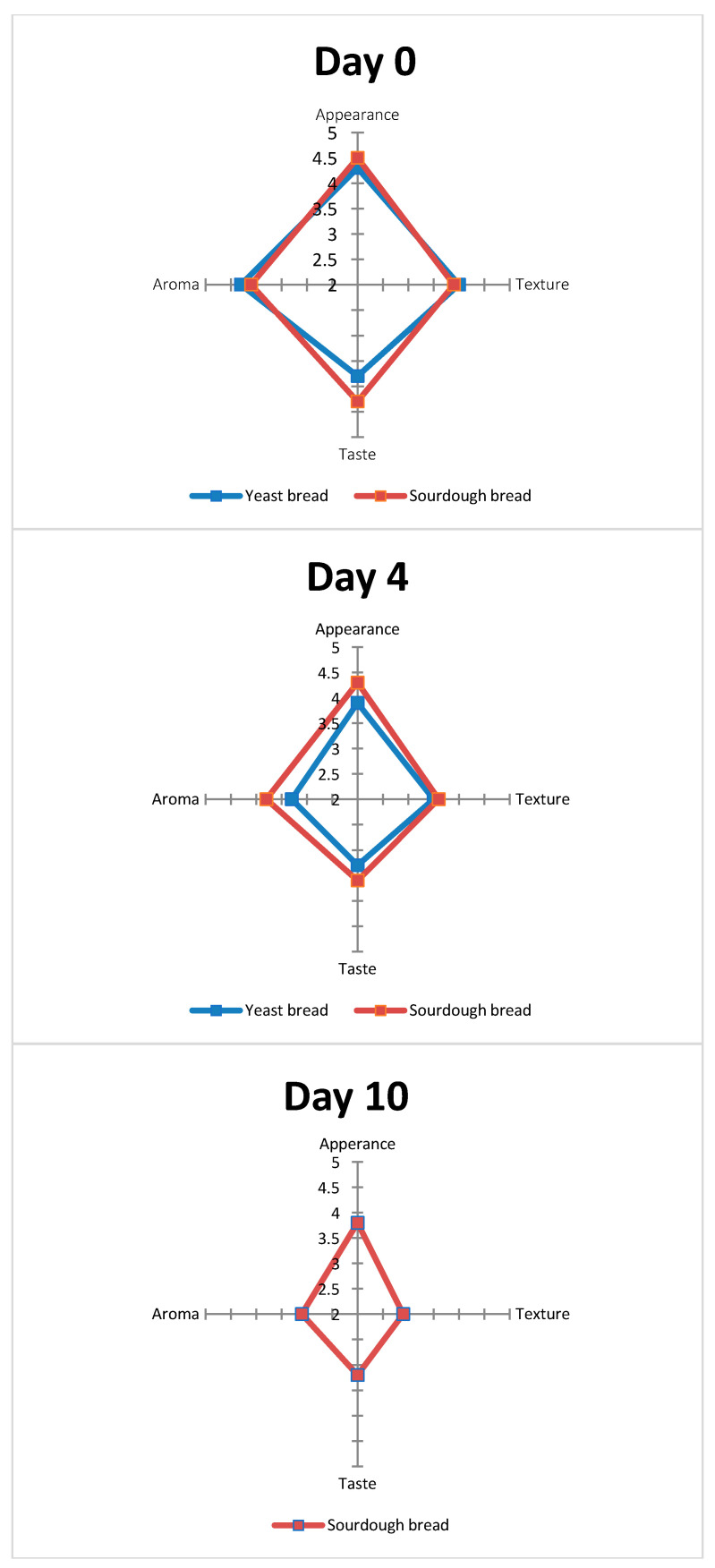
Sensory evaluation of yeast and sourdough breads in the form of star diagrams at different storage times.

**Table 1 foods-10-00635-t001:** Number of raw and quality filtered reads, number of sequences after chimera removal, and number of identified operational taxonomic units (OTUs) for 16S rRNA, 18S rRNA, and internal transcribed spacer (ITS) amplicon sequencing.

Amplicon	Number of Raw Reads	Number of Reads after Quality Filtering	Number of Reads after Chimera Filtering	Number of OTUs Identified
16S	107.167	72.435	70.145	61
18S	143.801	92.318	88.189	13
ITS	76.328	42.361	14.139	75

**Table 2 foods-10-00635-t002:** Effect of storage time on evolution of volatile (μg/kg) in sourdough bread stored at 25 °C.

	Day 0	Day 4	Day 8	Day 12	Day 16	Day 18	KIEx	KILt
Alcohols								
Ethanol	335.50 ± 1.28 c	343.40 ± 2.20 c	219.30 ± 3.50 b	214.70 ± 2.96 b	212.80 ± 3.08 b	190.50 ± 7.40 a	<500	<500
2-Methyl-1-propanol	6.13 ± 0.07 c	5.33 ± 0.43 bc	4.74 ± 0.12 ab	4.20 ± 0.15 a	4.24 ± 0.58 a	4.16 ± 0.11 a	623	625
3-Methyl-1-butanol	73.45 ± 1.00 a	69.69 ± 5.87 a	47.49 ± 3.83 b	40.79 ± 0.41 ab	38.22 ± 0.25 a	38.11 ± 0.16 a	734	743
2-Methyl-1-butanol	14.20 ± 2.10 d	12.86 ± 1.36 d	9.54 ± 0.59 a	7.25 ± 0.22 a	8.28 ± 0.31 a	7.79 ± 0.25 a	738	748
1-Pentanol	7.01 ± 0.16 b	6.70 ± 0.39 b	4.69 ± 0.23 a	4.67 ± 0.23 a	4.50 ± 0.06 a	3.86 ± 0.82 a	765	766
1-Hexanol	28.44 ± 1.66 c	28.03 ± 2.01 c	18.68 ± 1.97 ab	19.46 ± 1.01 ab	21.42 ± 1.29 b	15.80 ± 0.56 a	867	862
1-Octen-3-ol	n.d.	n.d.	n.d.	n.d.	n.d.	57.53 ± 30.10 b	981	978
Sum	464.73 ± 1.28	466.01 ± 2.75	304.74 ± 2.28	291.07 ± 1.30	289.46 ± 1.40	317.75 ± 3.05		
Aldehydes								
3-Methyl-1-butanal	15.85 ± 0.44 d	4.03 ± 0.47 c	3.35 ± 0.01 bc	1.92 ± 0.44 a	2.56 ± 0.13 ab	2.90 ± 0.46 ab	656	650
Hexanal	26.63 ± 1.38 d	14.69 ± 2.65 c	11.28 ± 0.17 bc	10.86 ± 0.20 b	12.46 ± 0.45 bc	6.23 ± 1.17 a	802	798
Furfural	10.70 ± 1.60 c	9.78 ± 1.51 c	3.38 ± 0.11 b	1.08 ± 0.54 ab	n.d.	n.d.	840	831
Heptanal	8.61 ± 0.18 c	5.71 ± 1.35 b	2.80 ± 0.32 a	3.59 ± 0.24 a	2.99 ± 0.17 a	2.20 ± 0.27 a	904	901
Benzaldehyde	10.15 ± 0.87 c	9.39 ± 0.79 c	3.92 ± 1.86 b	6.58 ± 0.35 c	4.40 ± 0.79 bc	n.d.	980	970
Octabal	2.95 ± 0.43 c	2.90 ± 0.67 bc	1.60 ± 0.13 a	1.93 ± 0.12 ab	2.32 ± 0.36 abc	4.04 ± 0.06 d	1007	1004
Nonanal	n.d.	2.55 ± 0.39 ab	3.12 ± 0.29 b	9.42 ± 2.37 c	12.41 ± 1.03 d	n.d.	1109	1099
Sum	74.89 ± 0.96	49.05 ± 1.34	29.45 ± 0.72	35.28 ± 0.95	37.14 ± 0.59	15.37 ± 0.64		
Ketones								
2-Pentyl-furanone	3.15 ± 2.23 a	2.62 ± 0.61 a	1.18 ± 0.37 a	2.19 ± 0.04a	1.94 ± 0.02 a	6.61 ± 0.64 b	993	989
Esters								
Butanoic acid, methyl ester	4.36 ± 0.29 d	1.95 ± 0.21 a	2.12 ± 0.07 ab	2.61 ± 0.20 bc	2.09 ± 0.18 a	3.12 ± 0.08 c	721	735
Sum	4.36 ± 0.29	1.95 ± 0.21	2.12 ± 0.07	2.61 ± 0.20	2.09 ± 0.18	3.12 ± 0.08		
Organic Acids								
Acetic acid	115.90 ± 6.10 abc	99.30 ± 14.5 a	131.00 ± 6.68 bc	137.14 ± 5.90 c	122.90 ± 6.90 bc	113.24 ± 2.40 ab	581	608
Ηydrocarbons								
Heptane	6.09 ± 0.31 b	4.24 ± 0.09 a	4.17 ± 0.27 a	3.59 ± 0.04 a	3.87 ± 0.19 a	4.75 ± 1.28 ab	698	700
Nonane	2.37 ± 0.09 c	1.45 ± 0.08 b	1.20 ± 0.05 ab	1.49 ± 0.27 b	0.86 ± 0.04 a	2.55 ± 0.07 c	901	900
Decane	n.d.	3.02 ± 0.49 ab	1.70 ± 0.62 b	3.29 ± 0.16 c	2.05 ± 0.02 bc	32.14 ± 0.95 d	1001	1000
Dodecane	n.d.	n.d.	0.88 ± 0.44 a	n.d.	n.d.	9.82 ± 0.84 b	1202	1200
Tetradecane	n.d.	n.d.	3.51 ± 0.51 c	1.86 ± 0.37 b	n.d.	n.d.	1403	1400
dl-Limonene	1.30 ± 0.35 a	1.22 ± 0.35 a	0.97 ± 0.01 a	1.40 ± 0.27 a	4.87 ± 0.81 b	3.78 ± 0.43 b	1043	1039
gamma-Terpinene	4.33 ± 1.69 c	2.20 ± 0.62 b	n.d.	n.d.	n.d.	n.d.	1070	1062
Sum	14.09 ± 0.88	12.13 ± 0.39	12.43 ± 0.39	11.63 ± 0.25	11.65 ± 0.42	53.04 ± 0.82		
Aromatic compounds								
2-Pentyl-furan	19.82 ± 0.22 d	12.85 ± 1.74 c	5.08 ± 0.29 a	5.26 ± 0.26 a	4.10 ± 0.34 a	7.99 ± 0.77 b	994	992
p-Cymene	2.97 ± 0.58 c	2.71 ± 0.70 bc	1.89 ± 0.23 abc	1.82 ± 0.25 ab	0.85 ± 0.11 a	10.70 ± 0.29 d	1037	1026
Sum	22.79 ± 0.44	15.56 ± 1.32	6.79 ± 0.26	7.08 ± 0.25	4.95 ± 0.25	18.69 ± 0.58		
Sulfur compounds								
Dimethyl-disulfide	n.d.	n.d.	0.62 ± 0.44 b	n.d.	n.d.	0.46 ± 0.32 ab	751	746

n.d. = not detected. The values are the means of 6 measurements, (n = 6) ± S.D., a.b.c Means with different letters in the same row are statistically different (*p* < 0.05), KIEx = Kovac Index experimentally determined data, KILi = Kovac Index literature data, Nist 05, J. Wiley & Sons Ltd., West Sussex, England.

**Table 3 foods-10-00635-t003:** Effect of storage time on evolution of volatile (μg/kg) in yeast leavened bread stored at 25 °C.

Compounds	Day 0	Day 2	Day 4	Day 6	Day 8	KIEx	KILt
Alcohols							
Ethanol	447.60 ± 6.30 d	462.20 ± 15.97 d	312.43 ± 2.96 c	274.20 ± 7.63 b	11.86 ± 0.63 a	-	-
1-Propanol	1.42 ± 0.36 b	1.42 ± 0.27 b	1.19 ± 0.04 b	n.d.	n.d.	552	554
2-Methyl-1-propanol	37.26 ± 1.37 c	35.49 ± 2.35 c	22.98 ± 2.49 b	22.18 ± 1.38 b	14.02 ± 0.59 a	623	625
3-Methyl-1-butanol	169.70 ± 5.97 d	169.60 ± 5.72 d	90.09 ± 4.55 c	67.98 ± 2.49 b	3.87 ± 0.18 a	734	743
2-Methyl-1-butanol	88.46 ± 2.69 d	89.62 ± 3.11 d	46.86 ± 5.95 c	37.29 ± 0.11 b	7.86 ± 0.69 a	738	748
1-Pentanol	4.01 ± 0.65 c	4.11 ± 0.65 c	2.36 ± 0.17 b	n.d.	n.d.	765	766
1-Hexanol	18.29 ± 0.04 d	17.31 ± 1.07 d	9.97 ± 2.70 c	4.28 ± 0.83 b	n.d.	867	862
1-Octen-3-ol	n.d.	n.d.	4.70 ± 2.00 b	n.d.	88.35 ± 2.42 c	981	978
Sum	766.74 ± 3.48	779.75 ± 6.60	490.58 ± 3.21	405.93 ± 3.67	125.96 ± 1.19		
Aldehydes							
3-Methyl-1-butanal	2.49 ± 0.19 c	1.24 ± 0.09 b	0.98 ± 0.16 b	n.d.	2.77 ± 0.10 c	656	650
Hexanal	15.04 ± 0.51 d	11.74 ± 0.32 c	1.27 ± 0.51 b	n.d.	n.d.	802	798
Heptanal	6.94 ± 0.45 b	7.29 ± 0.91 b	0.46 ± 0.02 a	n.d.	n.d.	904	901
Benzaldehyde	4.12 ± 0.76 a	5.22 ± 1.48 a	n.d.	n.d.	n.d.	980	970
Octanal	4.03 ± 0.73 d	2.68 ± 0.09 cd	2.06 ± 0.37 bc	1.16 ± 0.82 ab	n.d.	1007	1004
Nonanal	3.16 ± 0.26 b	6.82 ± 0.35 c	n.d.	n.d.	n.d.	1109	1099
Sum	35.78 ± 0.52	34.99 ± 0.74	4.77 ± 0.33	1.16 ± 0.82	2.77 ± 0.10		
Ketones							
2,3-Butanedione	3.94 ± 0.28 b	4.31 ± 0.30 b	4.16 ± 0.41 b	5.67 ± 0.32 c	1.84 ± 0.42 a	584	584
3-Hydroxy-2-butanone	3.68 ± 0.17 b	3.41 ± 0.38 b	3.34 ± 0.41 b	3.11 ± 0.07 b	n.d.	710	708
2-Pentyl-furanone	3.12 ± 0.09 ab	2.76 ± 0.05 b	3.61 ± 0.51 c	2.78 ± 0.11 b	n.d.	993	989
Sum	10.74 ± 0.20	10.48 ± 0.28	11.11 ± 0.45	11.56 ± 0.20	1.84 ± 0.42		
Esters							
Formic acid, methyl ester	n.d.	n.d.	n.d.	n.d.	18.78 ± 0.39 b	-	-
Acetic acid, ethyl ester	n.d.	2.04 ± 1.04 b	0.43 ± 0.30 a	n.d.	n.d.	610	614
Butanoic acid, methyl ester	3.39 ± 0.52 ab	2.84 ± 0.36 b	4.13 ± 0.70 c	2.70 ± 0.10 b	1.41 ± 0.08 a	721	735
Sum	3.39 ± 0.52	4.88 ± 0.78	4.56 ± 0.34	2.70 ± 0.10	20.19 ± 0.28		
Hydrocarbons							
Hexane	4.04 ± 0.07 a	3.78 ± 0.24 a	3.92 ± 0.32 a	6.12 ± 0.12 c	5.54 ± 0.11 b	594	600
Heptane	6.72 ± 0.52 c	5.12 ± 0.06 ab	4.79 ± 0.11 a	5.83 ± 0.28 b	4.52 ± 0.27 a	698	700
Octane	n.d.	n.d.	0.78 ± 0.03 b	1.01 ± 0.07 c	1.67 ± 0.00 d	801	800
Nonane	2.94 ± 0.30 c	1.94 ± 0.20 b	1.42 ± 0.19 b	1.34 ±0.12 ab	0.77 ± 0.27 a	901	900
Decane	5.12 ± 0.30 b	3.56 ± 0.18 a	4.77 ± 0.40 b	3.39 ± 0.08 a	6.14 ± 0.04 c	1001	1000
Dodecane	6.16 ± 0.61 b	12.22 ± 0.77 c	11.19 ± 0.99 c	6.43 ± 0.45 b	2.82 ± 0.09 a	1202	1200
Tetradecane	1.92 ± 0.02 a	14.73 ± 2.60 c	20.4 ± 1.03 d	15.31 ± 1.21 c	7.17 ± 1.80 b	1403	1400
dl-Limonene	3.29 ± 0.15 b	2.93 ± 0.53 b	2.18 ± 0.01 a	4.68 ± 0.04 c	2.12 ± 0.07 a	1043	1039
Caryophyllene	n.d.	n.d.	n.d.	n.d.	5.49 ± 1.04 b	1459	1440
gamma-Terpinene	n.d.	n.d.	n.d.	n.d.	3.08 ± 0.78 b	1070	1062
Sum	30.19 ± 0.34	44.28 ± 1.05	49.45 ± 0.54	44.11 ± 0.47	39.32 ± 0.71		
Aromatic compounds							
2-Pentyl-furan	7.70 ± 0.61 c	4.28 ± 0.20 b	3.66 ± 0.53 b	n.d.	n.d.	994	992
p-Cymene	4.01 ± 0.09 ab	10.29 ± 6.28 b	3.18 ± 0.29 ab	2.65 ± 0.29 a	3.20 ± 0.31 ab	1037	1026
Sum	11.71 ± 0.44	14.56 ± 4.44	6.84 ± 0.43	2.65 ± 0.29	3.20 ± 0.31		
Sulfur compounds							
Dimethyl-disulfide	n.d.	n.d.	2.70 ± 0.09 c	1.34 ± 0.04 b	n.d.	751	746

n.d. = not detected. The values are the means of 6 measurements, (n = 6) ± S.D., a.b.c Means with different letters in the same row are statistically different (*p* < 0.05), KIEx = Kovac Index experimentally determined data, KILi = Kovac Index literature data, Nist 05, J. Wiley & Sons Ltd., West Sussex, England.

**Table 4 foods-10-00635-t004:** The effect of storage time on (a) hardness and (b) springiness development of yeast and sourdough bread stored at 25 °C.

a. Hardness				b. Springiness			
Yeast Bread		Sourdough Bread		Yeast Bread		Sourdough Bread	
Day	(N)	Day	(N)	Day	(mm)	Day	(mm)
0	4.06 ± 0.59 a	0	7.15 ± 1.94 a	0	3.76 ± 0.92a	0	1.74 ± 0.39 a
2	4.76 ± 2.58 ab	2	18.37 ± 1.88 ab	2	4.53 ± 0.85a	2	1.91 ± 0.60 a
4	11.78 ± 3.53 bc	4	26.21 ± 6.84 abc	4	5.07 ± 0.70a	4	2.00 ± 0.15 ab
6	11.88 ± 3.42 bc	6	49.34 ± 11.42 cd	6	3.42 ± 0.62a	6	2.07 ± 0.12 ab
8	12.52 ± 3.02 c	8	39.43 ± 7.67 bcd	8	4.72 ± 0.51a	8	2.61 ± 0.41 ab
		10	35.40 ± 5.89 bcd			10	3.28 ± 0.30 b
		12	53.78 ± 10.07 d			12	2.51 ± 0.49 ab
		14	52.65 ± 10.68 cd			14	2.93 ± 0.36 ab
		16	58.60 ± 10.83 d			16	2.76 ± 0.35 ab
		18	48.09 ± 16.20 cd			18	2.21 ± 0.61 ab

a, b, c, d Means with different letters in the same column are statistically different (Tukey’s test, *p* < 0.05).

## Data Availability

Not applicable.

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
