# Peer review of "Characterization of Artisanal Spontaneous Sourdough Wheat Bread from Central Greece: Evaluation of Physico-Chemical, Microbiological, and Sensory Properties in Relation to Conventional Yeast Leavened Wheat Bread"

_foods, 2021, doi:10.3390/foods10030635_

Round 1

Reviewer 1 Report

The authors characterized artisanal spontaneous sourdough wheat bread from central Greece relative to conventional yeast leavened wheat bread by studying physico-chemical, microbiological, and sensory properties. They presented some interesting findings, but there are some major problems that should be addressed.

According to the sensory evaluation, yeast bread was unacceptable after 4-5 days, while sourdough bread was unacceptable after 10 days. The author should also evaluate the shelf-life from the microbiological perspective using some microbiological standards for foods based on the microbial counts, since all microbial data are available. Most of the quality properties were measured without considering the shelf-life, which is not meaningful or logical. However, all properties should be investigated within the period of shelf-life.

The data should be interpreted more statistically to show if there were significant differences. P<0.05 or P>0.05 should be added when describing the data, although they were noted at some points.

Author Response

Comment 1. According to the sensory evaluation, yeast bread was unacceptable after 4-5 days, while sourdough bread was unacceptable after 10 days. The author should also evaluate the shelf-life from the microbiological perspective using some microbiological standards for foods based on the microbial counts, since all microbial data are available. Most of the quality properties were measured without considering the shelf-life, which is not meaningful or logical. However, all properties should be investigated within the period of shelf-life.

Response: shelf life of breads was considered from the microbiological perspective based on legal standards.  See revised text l. 62-66. To the best of our knowledge, there are no official legal standards  for physico-chemical parameters determined.

Comment 2.The data should be interpreted more statistically to show if there were significant differences. P<0.05 or P>0.05 should be added when describing the data, although they were noted at some points.

Response: statistical significant differences (p<0.05) were added where needed.

Reviewer 2 Report

The manuscript aims to evaluate the technological and sensory properties of a sourdough wheat bread and a yeast based leavening wheat bread.

I am frankly missing which is the real aim of this study. Authors seem to compare – but it is not done for all the measured markers – the sourdough preparation with the one made with yeast. I think this is not a useful approach, as these kind of preparation are totally different between them, so it is like comparing apples and pears. It would have made sense if authors put some different variables in the preparation of the two single items and then compare the markers.

It is well known that bacteria strains are different in these two preparations, so nothing has been added to the existing literature.

Author Response

Comment. I am frankly missing which is the real aim of this study. Authors seem to compare – but it is not done for all the measured markers – the sourdough preparation with the one made with yeast. I think this is not a useful approach, as these kind of preparation are totally different between them, so it is like comparing apples and pears. It would have made sense if authors put some different variables in the preparation of the two single items and then compare the markers. It is well known that bacteria strains are different in these two preparations, so nothing has been added to the existing literature.

Response: As the reviewer notes the different variable included in the present study between yeast leavened and sourdough bread is that of the starter cultures used in breadmaking. In yeast leavened bread, baker’s yeast consists of Saccharomyces cerevisiae. In the sourdough bread, the starter culture is a combination of Lactic acid bacteria and yeasts, quite different than that of conventional bread. Thus, the present study investigates the qualitative and quantitative effects of different starter cultures (specific cultures used in this part of Greece) on the physicochemical, microbiological and sensory properties of bread. New information provided is the effect of particular starter cultures used on bread quality. In our opinion we are comparing “pears to apples” but bread to bread. 

Reviewer 3 Report

The submitted manuscript describes the study in which the sourdough wheat bread and yeast leavened wheat bread were compared in terms of multiple factors (microbiological,  physicochemical and sensory). The study is well designed, described and concluded. My comments are minor ones.

Line 27, “and” should not be written using italics.

Line 73, please define “shelf life of bread”.

Figure 1, why the initial TVC of yeast bread is significant, despite, as the Authors write (line 221) “Bread is considered a sterile product being heated at a high temperature during baking”

Line 424, not “hydroscopic” but “hygroscopic”.

Line 434: A comma after aldehydes is missing.

Table 2, values are not rounded correctly, the uncertainty should be rounded off to one or two

significant figures; besides: nonanal, day 4, shouldn’t it be just “a”?

Table 4, values are not rounded correctly, the uncertainty should be rounded off to one or two

significant figures

Figure 6, day 10, figure legend is missing.

It would be nice if the Authors end the conclusions section with the single phrase comparing sourdough wheat bread and yeast leavened wheat bread.

Author Response

1.Line 27, “and” should not be written using italics.

Response: done

2.Line 73, please define “shelf life of bread”.

Response: to the best of our knowledge there is no such definition. By ‘bread shelf life’ we refer to that time period during which bread will retain its desirable physico-chemical, microbiological and sensory characteristics.

3.Figure 1, why the initial TVC of yeast bread is significant, despite, as the Authors write (line 221) “Bread is considered a sterile product being heated at a high temperature during baking”

Response: we never stated that the initial TVC of bread is especially significant. We have recorded this value to show the evolution of TVC during storage.

4.Line 424, not “hydroscopic” but “hygroscopic”.

Response: done

5.Line 434: A comma after aldehydes is missing.

Response: done

6.Table 2, values are not rounded correctly, the uncertainty should be rounded off to one or two

Response: all values have been rounded  off to 2 decimal points

7.significant figures; besides: nonanal, day 4, shouldn’t it be just “a”?

Response: the comment is logical. The statistical program, has taken the nonanal non detected sample as superscript ‘a’. Thus the value 2.55 lies between 0.00 and 3.12  resulting in a superscript of ‘ab’. 

8.Table 4, values are not rounded correctly, the uncertainty should be rounded off to one or two

Response: done

9.Figure 6, day 10, figure legend is missing.

Response: legend for Fig. 6, Day 10 has been included.

10.It would be nice if the Authors end the conclusions section with the single phrase comparing sourdough wheat bread and yeast leavened wheat bread.

Response: text in the conclusion section has been rearranged to satisfy the reviewer’s comment.

Round 2

Reviewer 1 Report

You should first determine the shelf-life from the sensorial and microbiological perspectives. And then, all properties should be investigated within the period of shelf-life. It doesn’t make sense if you evaluated the quality properties if the samples were not within the shelf-life.

Author Response

Reviewer’s comment:

‘You should first determine the shelf-life from the sensorial and microbiological perspectives. And then, all properties should be investigated within the period of shelf-life. It doesn’t make sense if you evaluated the quality properties if the samples were not within the shelf-life’.

Response:

When one collects data within a specified time period it is only logical to comment on all findings. Then, based on legal standards for microbiological data and sensory evaluation findings one should focus on quality parameter values that determine actual product shelf life. This is what we have done.

Reviewer 2 Report

The manuscript has been reviewed by authors. I still have the same comments made in the past revision, as authors just commented my comments.

In yeast leavened bread, baker’s yeast consists of Saccharomyces cerevisiae. In the sourdough bread, the starter culture is a combination of Lactic acid bacteria and yeasts, quite different than that of conventional bread.

I thank authors for the kindly explication about the difference. As they note, one is a yeast and the other is a bacteria. So, despite they are breads, the fermentative part is completely different, from this my expression "comparing apple to pears".

Thus, the present study investigates the qualitative and quantitative effects of different starter cultures (specific cultures used in this part of Greece) on the physicochemical, microbiological and sensory properties of bread.

This is not true. Authors characterized the sourdogh matrix for the bacteria present, they did not investigate any difference in this starter culture, because it cannot be compared any metabolite from this culture and the Saccharomyces one. As said in my previous comment, if they investigated any difference within these two categories of products, that is investigating differences.

Author Response

The manuscript has been reviewed by authors. I still have the same comments made in the past revision, as authors just commented my comments.

In yeast leavened bread, baker’s yeast consists of Saccharomyces cerevisiae. In the sourdough bread, the starter culture is a combination of Lactic acid bacteria and yeasts, quite different than that of conventional bread.

That is, our study focused on (i) how the use of baker’s yeast alone (Saccharomyces cerevisiae) vs. sourdough (lactic acid bacteria + specific yeasts) affected bread quality characteristics including bread shelf life and (ii) identification of specific sourdough microbiota. 

I thank authors for the kindly explication about the difference. As they note, one is a yeast and the other is a bacteria. So, despite they are breads, the fermentative part is completely different, from this my expression "comparing apple to pears".

Thus, the present study investigates the qualitative and quantitative effects of different starter cultures (specific cultures used in this part of Greece) on the physicochemical, microbiological and sensory properties of bread.

This is not true. Authors characterized the sourdogh matrix for the bacteria present, they did not investigate any difference in this starter culture, because it cannot be compared any metabolite from this culture and the Saccharomyces one. As said in my previous comment, if they investigated any difference within these two categories of products, that is investigating differences.